# Nested Real-Time PCR Assessment of Vertical Transmission of Sandalwood Spike Phytoplasma (‘*Ca*. Phytoplasma asteris’)

**DOI:** 10.3390/biology11101494

**Published:** 2022-10-12

**Authors:** Kiran Kirdat, Bhavesh Tiwarekar, Purushotham Swetha, Sodaliyandi Padma, Vipool Thorat, Kathiruguppe Nagappa Manjula, Narayan Kavya, Ramachandran Sundararaj, Amit Yadav

**Affiliations:** 1National Centre for Cell Science, NCCS Complex, SP Pune University Campus, Ganeshkhind, Pune 411007, India; 2Forest Protection Division, Institute of Wood Science & Technology, 18th Cross, Malleswaram, Bengaluru 560003, India

**Keywords:** Sandalwood Spike disease, phytoplasma, seed transmission, real-time PCR

## Abstract

**Simple Summary:**

The Sandalwood Spike disease (SSD) related to ‘*Ca*. Phytoplasma asteris’ has almost wiped out the sandalwood population from the forests of southern India. It is known that sap-sucking insect vectors transmit phytoplasmas; however, their transmission through seeds needs thorough investigation. We found that 38.66% and 23.23% of one-month and four-month-old seedlings, respectively, tested positive for SSD phytoplasma screened using modified real-time qPCR assays in insect-free environments. Considering the current efforts to reestablish the healthy sandalwood population and its commercial importance, these findings are worrisome. The role of some other microbes in the high mortality rates of sandalwood seedlings remains unknown and requires further investigation.

**Abstract:**

The Sandalwood Spike disease (SSD)-related to ‘*Ca*. Phytoplasma asteris’ has threatened the existence of sandalwood in India. The epidemiology of SSD is still poorly understood despite the efforts to understand the involvement of insect vectors in SSD transmission and alternate plant hosts over the last two decades. Apart from the transmission of SSD phytoplasma through insect vectors, the information on vertical transmission is entirely unknown. Over 200 seeds from SSD-affected trees and over 500 seedlings generated using commercially purchased seeds were screened for the presence of SSD phytoplasma to understand the vertical transmission in an insect-free environment. The end-point nested PCR and real-time nested PCR-based screening revealed an alarming rate of 38.66% and 23.23% phytoplasma positivity in one-month and four-month-old seedlings, respectively. These results were further validated by visualizing the phytoplasma bodies in sandalwood tissues using scanning electron microscopy. The presence of phytoplasma DNA in the seeds and seedlings is a concern for the commercial distribution of sandalwood seedlings in the current setup. This also poses a fear of spreading the disease to newer areas and negatively affecting the economy. The seedling mortality was also suspected to be associated with isolated bacterial and fungal isolates such as *Erwinia, Curtobacterium*, *Pseudomonas, Rhodococcus*, *Aspergillus*, *Fusarium*, and *Neofusicoccum* isolated using a culture-dependent approach. These findings strongly recommend the accreditation of commercial production of sandalwood seedlings curtailing SSD phytoplasma’s menace. Additionally, a new nested end-point and qRT PCR assays developed in this study proved valuable for the rapid screening of phytoplasma in many plant samples to detect phytoplasmas.

## 1. Introduction

The Indian sandalwood (*Santalum album* Linn. Family: *Santalaceae*) is the second most expensive wood in the world. The sandalwood tree is treasured due to the aromatic essential oil derived from the heartwood and makes an exquisite natural material for carvings. India was once a world leader in sandalwood heartwood production, fulfilling 80% of global annual demand with over 4000 tonnes of production in the 1960s [1,2]. The Sandal spike disease (SSD) has adversely affected the natural population of sandalwood in the southern states of India (the parts of mainly Karnataka and Tamilnadu, and Kerala states), leading to a global shortage of sandalwood-based products [3,4]. The sandalwood production in India has been decreasing annually at a rate of 20 percent since 1995. From 1997–1998, a total of 27,930 kg of sandalwood oil was exported, which was reduced to 10 Kg in 2015–2016 [4,5]. The decline in natural population pushed *S. album* into the International Union for Conservation of Nature (IUCN) ‘red list’ of threatened species in 1998 and listed as ‘vulnerable (*vu*)’ in 2019 [6]. The SSD is related to the endophytic and obligate bacterial pathogen, ‘*Candidatus* Phytoplasma’ [7]. This pathogen is associated with thousands of crops, weeds, and wild plants and is responsible for extensive yearly yield losses [8]. Seldom attempts have been made to obtain an axenic culture of the ‘*Ca*. Phytoplasma’, however, was not convincing [9,10,11]. The phytoplasma infection in sandalwood results in symptoms such as chlorosis, leaf yellowing, reduction in leaf size, and stems standing out stiffly with a spike-like appearance due to crowded leaves. This causes the dieback of branches leading to the death of young and mature sandalwood trees [4,12]. The SSD phytoplasmas are phylogenetically related to the aster yellows (16SrI-B) group belonging to ‘*Ca*. Phytoplasma asteris’ [13,14,15]. In 2020, mixed phytoplasma infections of Sugarcane Grassy Shoot (SCGS) phytoplasma (‘*Ca*. P. sacchari’, 16SrXI-B group) and aster yellows phytoplasma were observed in Marayoor Sandalwood Reserve (Kerala state) region [4]. The genome of SSD phytoplasma (JAGVSK000000000) has been sequenced recently, enhancing the understanding of genetics and characteristics related to pathogenesis.

The SSD has been known for over 100 years, and numerous epidemiological studies were performed to understand the causal agent, transmission, and development of the disease. The sap-sucking insect vectors transmit phytoplasmas, and so far, *Nephotettix virescens*, *Moonia albimaculata*, and *Coelidea indica* were inferred as putative vectors [16]. However, a complete epidemiological characterization of the SSD phytoplasma insect vectors is lacking to suffice the need for managing the disease. The vertical transmission of phytoplasma through seeds has been a debatable issue over past years, considering the poor vascular connection of the embryo with the mother plant. However, phytoplasma DNA was detected in companion cells of *Oenothera elata* sp. hookeri [17] and parenchyma cells of *Cuscuta odorata* [18]. In a controlled growth environment, the seedlings of oilseed rape, tomato, and maize carried the same phytoplasma detected in the infected mother plants [19]. Similar results were reported in the case of maize (*Zea mays*), where identical phytoplasma (‘*Ca.* Phytoplasma cynodontis’) was detected in plant tissue, their respective seeds, and seedlings grown in insect-free cages and tested in different batches [20]. The 4-month-old carrot seedlings were reported to possess phytoplasma when grown in insect-free cages [21]. However, the studies carried out with West African Tall coconut palms reported no phytoplasma in seedlings, although phytoplasma DNA was detected in the inflorescences and embryos [22]. So far, no studies have been performed to check the phytoplasma spread through sandalwood seeds, given the high incidence of SSD.

In India, the nursery-raised seedlings have been a primary source of sandalwood saplings distributed to the private and state forest departments. These seedlings are primarily raised by the seeds obtained from the sandalwood trees growing in the Marayoor Sandalwood Reserve (MSR) situated in the southern state of Kerala. The MSR is known for hosting a large population of sandalwood trees free from SSD infestation [23] compared to other sandalwood growing areas, therefore, it is trusted in the seed market. This study was initiated after finding the phytoplasma positive seedlings, raised using seeds from MSR, collected randomly from the commercial nurseries, and reports of the unknown reason for their mortality. The experiments were then designed to validate the seed-borne vertical transmission of SSD phytoplasma through commercially available seeds and seedlings produced using them. The sandalwood seedlings and seed tissues were screened for the presence of the pathogen. Here, we report the nested real-time PCR (qPCR)- assessment of the vertical transmission of the SSD phytoplasma through the seedlings grown using commercially available seeds and its significance in managing the SSD in nursery settings.

## 2. Materials and Methods

### 2.1. Source of Sandalwood Seeds and Seedlings

Ripened, purple-colored, single-seeded (drupe) sandalwood fruits (*n* = 226) were collected from SSD-affected trees in Male Mahadeshwara Hills, private cultivation at Chikmagaluru, and Chamundi Hills Karnataka (India), in multiple batches. The fruit samples were collected from symptomatic trees only. The individual fruit was washed and de-pulped within 48 h post harvesting to avoid secondary infections by gentle manual squeezing, avoiding any damage to the seed coat (Figure 1). Simultaneously, sandalwood seeds (*n* = 900), assumptively collected from asymptomatic sandalwood trees, were purchased at Marayoor, Kerala (India) in multiple batches from December 2020 to December 2021.

The commercially purchased and de-pulped seeds were washed thoroughly and immersed in sterile water. The floating seeds were discarded, the immersed seeds were selected and shade dried for three days at room temperature (27 °C). Further, seeds were soaked in 1500 ppm gibberellic acid (GA3; Cat. No. PCT0830; Himedia, India) for 16 h to break the seed dormancy. These GA3-treated seeds were sown in the freshly autoclaved sand spread in a nursery bed (10 m × 1 m × 0.2 m) and mixed with nematicides (Quinalphos, 0.5 gm per kg of sand). The seedbeds were maintained in insect-proof greenhouse conditions, under shade, and kept moist by watering twice a day at room temperature. The germinated seedlings at the 2–4 leaflet stage were transplanted to polybags 30 to 45 days after sowing (DAS). The germination of sandalwood seedlings usually starts from 20 to 30 DAS. Seedlings were transferred to the potting mix containing sand, soil, compost, and nematicide filled in black polybags (2L) for a better spread of haustoria. Sandalwood being a semi-parasitic plant, *Alternanthera* sp. (Amaranthaceae) was used as a host plant during transplantation (Figure 1d). Fungicide (Bavistin, 1 gm/L) was sprayed at a regular interval of 15 days to keep the fungal infections under control. An appropriate amount of NPK fertilizers were used to improve the growth of seedlings.

All seeds (*n* = 226) collected in multiple batches from SSD symptomatic plants were screened for the presence of SSD phytoplasma. The first batch of 41 seeds collected from symptomatic plants was sowed to obtain the seedlings. A total of nine germinated seedlings were screened for the presence of SSD phytoplasma. The screening of seedlings generated from seeds of symptomatic trees was limited to one batch of nine seedlings due to the high mortality rate of germinated seedlings. No other seeds were collected from SSD symptomatic plants to generate the seedlings.

The batches of seedlings generated using commercially obtained seeds were collected at different time points days after sowing. A total of 524 seeds of 900 seeds purchased were germinated in insect-proof greenhouse conditions. Of these 524 germinated seedlings, 261 symptomatic seedlings were sampled in multiple batches (Figure 2). These samples were collected at two different time points of days after sowing (DAS), viz. a total of 119 seedlings were sampled 30–40 DAS, and 142 were collected 100–120 DAS and stored at −80 °C until further processing.

A total of 51 leaf samples of symptomatic sandalwood plants showing typical spike symptoms: stunting, shortened internodes, chlorosis, extensive reduction in leaf size, and the intense proliferation of leaf and flower buds were collected from severely affected areas of Marayoor Sandal Reserve (Kerala) and private cultivation in Chikmagaluru (Karnataka) and treated as positive controls (PC) ((Figure 3). The leaves of asymptomatic sandalwood trees were collected, confirmed for phytoplasma absence, and used as negative controls.

### 2.2. DNA Extraction, Nested End-Point PCR, and Sequencing

To confirm the presence of phytoplasmas, the genomic DNA was extracted using a CTAB method from 100 mg of leaf tissue from symptomatic, asymptomatic plants, seeds, and seedlings [24]. All the seed and seedling samples were crushed to powder using liquid nitrogen. The seed coat was excluded while extracting the genomic DNA from sandalwood seeds.

The phytoplasma 16Sr RNA gene was amplified in nested end-point PCR assay using primers P1 [25] and P7 [26], followed by newly designed nested PCR primers viz. R16.100F (GACGAGGATAACCRTTGGAAAC) and R16.1386R (GGTATTGCCAACTTT CGTGG). The PCR reaction was performed using final concentrations of 1X PCR buffer, 1.5 mM MgCl_2_, 50 ng genomic DNA, 200 µM of dNTPs, 1 U of LA Taq DNA Polymerase (Cat. No., RR002, TAKARA, Shiga, Japan), and 1 µM primers. For nested PCR reaction, a 20-fold diluted template generated by P1/P7 primers was used. The PCR products were purified using the PEG-NaCl method [27]. Briefly, the PCR products were precipitated by adding an equal volume of PEG-NaCl solution (20%, Polyethylene glycol (PEG), Cat. No. 1546605, Merck, USA; 2.5 M, NaCl, Cat. No. S3014, Merck, Darmstadt, Germany) followed by two washes of 70% ethanol and reconstituted with nuclease-free water.

The purified products were sequenced directly using bacterial universal primers 343R, 704F, 907R, 1028F, and 1492R [28] on ABI^®^ 3730XL DNA Analyzer (Applied Biosystems, Waltham, MA, USA). The 16S rRNA gene sequences were assembled, curated manually, and analyzed using the EzTaxon database [29] to find the closest match.

### 2.3. Assessment of Vertical Transmission of SSD Phytoplasma Using qPCR

All collected samples were screened for the presence of phytoplasma using end-point nested PCR using P1/P7 primers followed by nested R16.100F and R16.1386R primers as described above. All samples were further screened with qPCR using a TaqMan probe (5′ TGACGGGACTCCGCACAAGCG) specific for phytoplasma 16S rRNA genes and forward primer (5′ CGTACGCAAGTATGAAACTTAAAGGA), Reverse primer (5′ TCTTCGAATTAAACAACATGATCCA) [30]. The isolated genomic DNA from various tissues was used directly as a template.

Additionally, the P1/P7 derived amplicons were provided as a template for nested qPCR analysis. These templates were obtained by purifying the P1/P7 derived amplicons by Agencourt SPRIselect beads (Cat. No. B23318, Beckman Coulter; Brea, CA, USA) as per manufacturer’s instructions and eluted in 15 μL TE buffer. A 75 bp purified amplicon obtained using the primers mentioned above was quantified, and serially diluted (10^–3^ to 10^–7^) to obtain the standard curve. The qPCR assays for templates viz. genomic DNA (gDNA) and P1/P7 derived purified template were identical and described below.

The qPCR assays were performed on the StepOne Plus PCR system (Applied Biosciences, Waltham, MA, USA) using Premix Ex Taq™ (Cat. No. RR390A, TAKARA, Shiga, Japan). Each triplicate reaction contained 25 ng of gDNA or 2 μL of P1/P7 derived purified PCR product (nested qPCR), 0.4 µM of the forward and reverse primers and 0.2 μM probe, and 2x Premix Ex Taq (Probe qPCR) Master Mix to a final volume of 10 μL. The qPCR assay started with initial denaturation of 1 min at 60 °C and was followed by 40 cycles of amplification at 15 s at 95 °C and 30 s at 60 °C. The baseline and fluorescence thresholds were automatically determined for each sample. The presence of phytoplasma in a sample was confirmed if the cycle threshold (Ct) value did not exceed 36. All assays from each plate contained a no template control (NTC), genomic DNA from the healthy plant as negative control (NC), a calibrator DNA of known concentration as a positive control (PC) for comparison across the assay plates, five standard dilutions (STD) of known concentrations, and 24 samples. All samples and controls were added in triplicate to the plate for the real-time PCR assays. The *t*-test was performed to establish the statistical significance of the difference in positivity percentage across the PCR methods used for screening sandalwood samples for phytoplasma presence.

### 2.4. Assessment of Phytoplasma Presence by Scanning Electron Microscopy

A total of four of each fruit, seedling hypocotyl tissue, and two leaf samples tested PCR positive for phytoplasma selected for scanning electron microscopy assessment. The selected plant samples, which were hand sliced to 0.5 to 1 mm thickness, were fixed in 2.5% glutaraldehyde solution (G5882; Merck, Darmstadt, Germany. The fixed samples were dehydrated using graded alcohol concentrations of 20%, 30%, 40%, 50%, 60%, 70%, 80%, and 90%, one time and twice in 100% for 10 min each [31]. The dehydrated samples were coated with gold particles using a sputter coater (model, SC7620; Quorum Technologies, Lewes, UK). The size and shape of phytoplasma and other endophytic microbial cells were visualized and determined by a scanning electron microscope (Carl Zeiss, EVO 18, Version 6.02).

### 2.5. Culture-Dependent Screening of Endophytic Microbes

Symptomatic seedlings (30–40 DAS) that tested negative for phytoplasma infection were subjected to further screening of bacterial and fungal pathogens. The collected seedlings (*n* = 14) were surface sterilized and crushed separately into PBS buffer, and the lysate was plated on Tryptic Soy Agar (Cat. No. M1968, Himedia; Mumbai, India) and Potato Dextrose Agar (Cat. No. M096, Himedia; Mumbai, India) media. The bacterial and fungal colonies were isolated, purified, and subjected to genomic DNA isolation. These isolates were identified by sequencing the 16S rRNA gene amplified using bacterial universal primers (27F and 1492R) [28] and fungal ITS region using universal primers ITS1 and ITS4 [32], respectively. The obtained sequences were assembled, curated manually, and analyzed using the EzTaxon and GenBank databases to find the closest matches. All obtained DNA sequences were deposited in EMBL/GenBank/DDBJ database.

## 3. Results

### 3.1. Detection of Phytoplasma in Symptomatic Plant and Seeds

The symptomatic sandalwood plant samples (PC) were screened for the presence of phytoplasma using nested end-point PCR tested positive and showed 49/51 (98%) positivity using R16.100F and R16.1386R primers. The obtained 16S rRNA gene sequences of the phytoplasma strains in these samples showed 99.5% to 99.92% similarity to ‘*Ca*. P. asteris’ strain OY-M belongs to the Aster Yellows (AY) or 16SrI -B group. The representative sequences were submitted to GenBank (OM855940 to OM855973).

The genomic DNA of these samples (PC) was subjected to qPCR as a template, where 37 of 51 (72.5%) samples were found positive with a mean Ct value of 21.48 and a standard deviation of 4.15 (Figure 4b and Figure 5). Further, the nested qPCR indicated that 98% of samples (49 of 51) were positive for phytoplasma with a mean Ct value of 16.56 and a standard deviation of 6.65 (Figure 4a and Figure 5). In both assays, no amplification was observed in no template control (NTC).

A total of 226 seeds (SymSD)were obtained from the fruits of SSD-affected trees, and the first batch of 41 seeds was sowed. Out of 41 seeds, only 9 showed germination (21.9%). Further, these germinated seeds showed no growth or development and eventually died. The association of phytoplasma was confirmed in these dead tissues. No seeds were sowed further, and the remaining 185 seeds (SymSD) were screened for the presence of phytoplasma, where 42 seeds (22.7%) were found positive using qPCR with an average Ct value of 24.41 with a standard deviation (SD) of 5.25. The nested qPCR could detect phytoplasma DNA in 65 seeds (35.14%) with an average Ct value of 21.99 ± 8.76 (Figure 4 and Figure 5).

### 3.2. Seed Germination, Symptoms, and Phytoplasma Detection

Out of 900 purchased and sowed seeds, 524 were germinated, accounting for about 58.2% germination rate. The seed germination was initiated after 20–25 days, and seedlings of two to four-leaf stages were seen within 35–40 days. Profuse fibrous secondary roots developed in sand medium, supporting plants’ early establishment and survival in polybags.

A total of 119 symptomatic seedlings were sampled at 35–40 DAS and screened for phytoplasma presence. The symptomatic seedlings showed symptoms of a little leaf, leaf curling, stunted slow growth with shortened height, multiple abnormal shoots, shortened internodes, malformation of leaf laminae, mottling, and puckering of leaves leading to drying and abscission of leaves (Figure 2). These seedlings showed 5.88% positivity (7/119) when screened using qPCR (gDNA as template) with a Ct range of 29.76 SD ±5.99. The positivity rate was 38.66 % (46/119) when these samples were screened using nested qPCR with a mean Ct of 25.14 SD ±4.28, while it was 11.76% (14/119) when tested using end-point nested PCR (Figure 4 and Figure 5).

The second batch of symptomatic seedlings (*n* = 142) was harvested at 100–120 DAS. The phytoplasma DNA was detected in 5 out of 142 screened seedlings (3.52%) in gDNA qPCR (Ct 27.95 ± 6.35), and 33 (23.24%) were found positive using the nested qPCR assay (Ct 25.08 ± 5.09) whereas end-point nested PCR yielded 15.49% (22/142) positivity rate (Figure 4 and Figure 5). Further, the t-test revealed the nested qPCR was the most sensitive technique (*p* = 0.0126, <0.05) to detect phytoplasma DNA considering the obtained Ct values across the screening methods.

The nested qPCR assays were carried out in triplicates following the MIQE guidelines, and distributed in 21, 96-well PCR plates. The PCR efficiency of all qPCR assays ranged from 99 to 110, and the regression values (R^2^) value remained close to 0.99 (Figure 6). The calibrator DNA used in all plates showed an average Ct of 20.31 with a standard deviation of 0.98, supporting the comparison of Ct values obtained across the plates.

### 3.3. Scanning Electron Microscopy

The presence of phytoplasma cells was visualized in seeds, hypocotyl tissue of seedlings, and leaf samples of SSD symptomatic trees. These cells were seen in sizes ranging from 200 nm to 800 nm and were irregular in shape (Figure 7). We found that the irregular size, shape, and random distribution of phytoplasma cells within a plant cell and tissue are their characteristic features. In the leaf and seedling hypocotyl tissue, the phytoplasma cells were found abundantly in palisade and spongy parenchyma but comparatively less in vascular bundles and ground tissue. In the fruit samples, the phytoplasma cells were abundant in the seed coat but absent in the fruit coat and mesocarp. The phytoplasma cells were present in the endocarp and endosperm tissue of the sandalwood seeds (Figure 7). The unknown endophytic bacterial cells were also detected in all types of sandalwood tissues (Figure 8). These cells ranged from 500 nm to 2600 nm and were definitive in their shape compared to phytoplasma cells.

### 3.4. Bacterial and Fungal Isolates

Symptomatic seedlings (30–40 DAS) that tested negative for phytoplasma infection were subjected to further screening of bacterial and fungal isolates. The isolated bacterial strains were *Rothia*, *Erwinia, Bacillus*, *Curtobacterium*, *Microbacterium*, *Rhodococcus*, and *Pseudomonas*, while fungal isolates were *Aspergillus*, *Colletotrichum*, *Fusarium*, and *Neofusicoccum*. The 16S rRNA sequences for bacterial and ITS sequences fungal strains were deposited in GenBank/EMBL/DDBJ database with accession numbers OM838448 to OM838470 (Appendix A) and OM838424 to OM838447 (Appendix A), respectively.

## 4. Discussion

The decline in sandalwood production is mainly due to the depletion of sandalwood trees in the forest due to Sandal Spike Disease (SSD), as deduced from several studies conducted over the last two decades [2,4,13,14,15,33,34,35]. Because of this, the production of disease-free sandalwood populations remained one of the focuses of managing the SSD. To address the susceptibility of sandalwood to SSD and its high incidence in the given area, new areas such as Marayoor were selected to raise the new populations. At present, reasonably healthy populations of sandalwood are available in Marayoor Sandalwood Reserve (MSR), Chamundi Hills, and Bannerghatta in Karnataka state. Since then, these areas have become the primary source of sandalwood seeds to raise new plantations [36]. The change in the policy of the Indian government in the decade 2000 facilitated the private cultivation of sandalwood in many Indian states. Therefore, many private sandalwood plantations started acting as seed sources out of commercial interests; these seeds are harvested from asymptomatic plants but are not necessarily pathogens-free. These ‘healthy-looking’ seeds, based on their size and appearance, are usually distributed among the farmers without any accreditation. The results presented in this study were initiated when seedlings collected from commercial nurseries tested positive for phytoplasma in a pilot study.

The seeds collected from SSD symptomatic plants showed the presence of phytoplasma DNA where all seed samples were devoid of seeds coat. As part of the maternal plant, seed coats may carry a pathogen. The corroborated results inferred from the nested PCR and nested qPCR assays confirmed that the seeds of symptomatic plants indeed contain phytoplasma DNA indicating the presence of phytoplasma cells. Further, the seeds collected from symptomatic plants showed a very low germination rate (9/41); the germinated seeds died with a high mortality rate (9/9). The high titer of phytoplasma in these seeds (Ct values ranging from 21.99 to 30.75 using nested qPCR, Figure 4) was likely responsible for high mortality in sandalwood seedlings at the early stages of growth. The screening of seedlings at 34–40 DAS supports this observation with a similar Ct range (Figure 4). The seeds collected from symptomatic plants showed a 35.14% (65/185) phytoplasma positivity rate (based on nested qPCR assays) which indicates the intermittent and arbitrary presence of SSD phytoplasma in these samples (Figure 5), likewise observed in *Euphorbia pulcherrima* [30].

The primary aim of the study was to establish the presence of viable phytoplasma in commercially available seeds and seedlings and avoid false positives arising from the presence of DNA from the dead cells. As expected, 900 seeds purchased for the experiments showed a 58.2% (524/900) germination rate, which ranges from 40 to 60%, with a seedling survival rate of 70 to 80% [37]. Among these grown seedlings, those screened at 34–40 and 100–120 DAS showed an alarming rate of 38.66% and 23.3% phytoplasma positivity (Figure 5). The high standard deviation in all tested sample groups suggests varied phytoplasma titer in tested seeds, seedlings, and positive controls (Figure 4). The seedlings at 100–120 DAS, which tested positive for phytoplasma, started showing the typical SSD symptoms indicating the presence of viable phytoplasma cells in the sandalwood seedlings (Figure 2). The uncultivable nature of phytoplasma poses a challenge the understanding the percentage of inoculum that remains viable. However, the results obtained in this study show that the phytoplasma cells survive the dormancy period and can resume the infection once the seeds are germinated. As the seedlings were maintained in insect-free cages under greenhouse conditions, the only possible source of phytoplasma in these seedlings is the viable inoculum carried by the seeds.

The positivity rate remained lowest in genomic DNA qPCR followed by nested end-point PCR and then in nested qPCR for sample groups. This difference is attributed to the sensitivity of the technique used where genomic DNA qPCR fails to detect phytoplasma DNA from low titer samples, such as in 34–40 DAS seedling samples (Figure 5). Nested qPCR outperforms genomic DNA qPCR and nested end-point PCR since it utilized a pre-amplified template (template generated using P1/P7 primers) and detected phytoplasma DNA at late Ct in earlier undetected samples. The nested qPCR assays produced sensitive, quick and reliable results as compared to conventional electrophoresis-based screening and assay controls used in these assays. Also, the least variation in the PCR efficiency and regression values across all 21-qPCR assay plates used supported the reliability of the assays. The standard DNA template obtained from PCR amplicon (75 bp) was used in all 21 plates along with other assay controls and showed the least variation in the Ct values for each dilution used (Figure 6). The *t*-test showed that the difference between Ct values is statistically significant (*p* = 0.0126, <0.05), and nested qPCR proves to be the most sensitive technique to detect phytoplasma. This extended analysis aimed to detect the lowest titer of phytoplasma present in any seed or seedling that may contribute to spreading SSD to a new area. Nested qPCR improves the sensitivity for detecting phytoplasma in various sandalwood tissue, seeds, and seedlings. This approach may be helpful for active SSD surveillance in areas where asymptomatic infections are prevalent, especially in commercial nurseries and conventional regions known for seed supply. The results obtained in this study used the seeds for sowing purchased from retailers in Marayoor and were assumptively collected from asymptomatic trees in MSR.

The TaqMan-based qPCR assays are sensitive and may detect other bacteria when plant metagenome DNA is used as a template [30]. We observed the inconsistent and insignificant appearance of the late Ct values in healthy plant genomic DNA control in a few wells during the pilot qPCR assays where they were most expected. Based on these observations, a cut-off value of 36 was set to avoid false positives. As far as nested PCR assay is concerned, we did not observe late Ct values (>36) where a qPCR assay used a template generated using P1/P7 primers (Figure 4).

Additionally, we modified the qPCR assay (called nested qPCR) designed based on the observation of the detection of other bacteria [30]. The exercise of using nested qPCR ensured the specificity of the template provided for nested qPCR compared to the genomic DNA. This methodology reduced the introduction of a high copy number of bacteria to almost negligible levels and provided a template for specific detection. Importantly, the study did not miss the true phytoplasma positives using nested qPCR due to a specific template, or it did not encounter false positives as no late Ct denoting bacterial presence in nested qPCR was recorded. The phytoplasma positive samples yielded comparable Ct values irrespective of the type of template used (Figure 4); this was also reported by earlier studies [30].

The presence of SSD phytoplasma in seeds and seedlings was confirmed by SEM (Figure 7). The intermittent and arbitrary presence of SSD phytoplasma in plant samples was validated again in this study. The presence of phytoplasmas in the seed endocarp and endosperm tissues (Figure 7) was a key finding of the study, which is most likely responsible for the vertical transmission further to seedlings. SEM imaging also proved to be less laborious in processing the samples for imaging, but although simple, SEM was time-consuming and at times confusing in recognizing the phytoplasma cells among the other endophytic microbiome present in the tissue sample (Figure 8).

The seedling samples tested negative for phytoplasma, showed abnormal growth were tested for the presence of other endophytic microbes. The presence of known endophytic bacterial and fungal strains indicated the poor physiological state of the seedlings. (Appendix A). Despite the seeds being treated with an anti-fungal agent and seedbeds being protected from external infections, the intrinsic endophytic inoculum of these organisms severely impacted the growth of the seedlings. The role of a number of microbes found needs to be investigated further for their role in high mortality rates in sandalwood seedlings for better management of nursery-raised seedlings and their production.

The transmission of phytoplasma cells through seeds has been a topic of discussion since Khan et al. (2002) demonstrated the presence of phytoplasma in alfalfa (*Medicago sativa*) seeds [38]. Satta et al. (2019) documented and discussed the studies related to phytoplasma transmission through seeds and other reproductive organs [39]. The discussion in these studies revolved around whether phytoplasma cells remain viable in the seeds for a long time, especially when phytoplasmas cannot be cultivated in vitro. Also, the seed anatomical structure shows the absence of a direct vascular connection between the phloem system and the embryos. Although many studies proved the presence of phytoplasma reproductive organs using electron microscopy [40,41], none demonstrated the possible movement of the phytoplasma cells including this study. Given this, this study demonstrated the presence of phytoplasma in sandalwood seeds and seedlings grown using them with a distributed percentage.

## 5. Conclusions

The presence of phytoplasma DNA in seeds and seedlings grown using them is alarming for commercial distribution with a fear of spreading to newer areas. The detection assays in this study indicate that the nested qPCR-based accreditation needs to be implemented to stop the spread of the SSD to new plantations. Further research is required to address the unanswered question of the movement of phytoplasma cells through tissues, especially those without vascular connections. The commensal microbiome associated with sandalwood seeds and seedlings needs further study of their role in seed and seedling physiology of growth and development.

## Figures and Tables

**Figure 1 biology-11-01494-f001:**
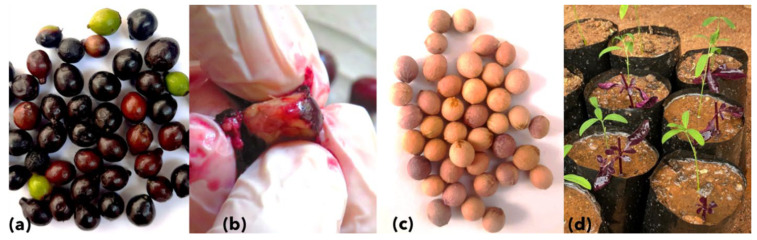
Sandalwood seed germination and seedlings generation. Collection of healthy drupe fruits (**a**), de-pulping of individual fruit (**b**), clean and dried seeds ready for sowing (**c**), and asymptomatic sandalwood seedlings at 25–35 DAS grown along with host plant *Alternanthera* sp. (Amaranthaceae). The host plant can be seen in deep purple color (**d**).

**Figure 2 biology-11-01494-f002:**
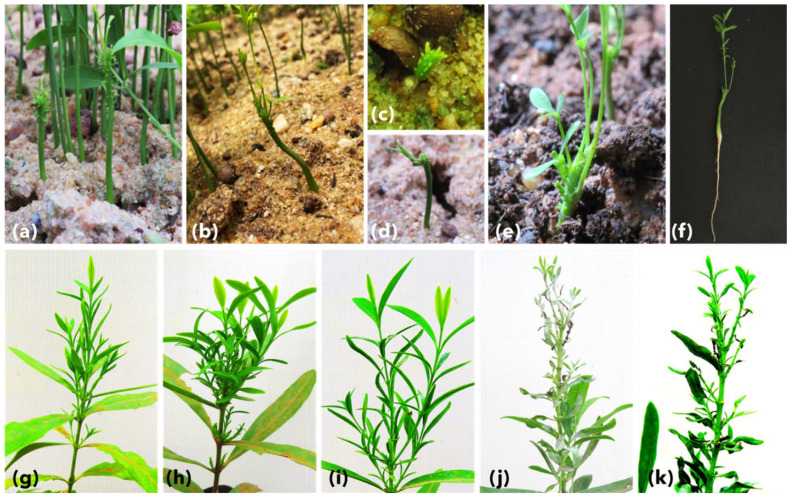
Sandalwood seedlings at 35–40 DAS showing SDD symptoms such as a little leaf, malformation of leaves and leaf laminae, stunted and abnormal growth, puckering of leaves, shortened internodes (**a**–**f**) and 100–120 DAS with SSD symptoms show symptoms of a little leaf, stunted growth, multiple abnormal shoots, shortened internodes, curling, mottling, and puckering of leaves leading to drying and abscission of leaves (**g**–**k**).

**Figure 3 biology-11-01494-f003:**
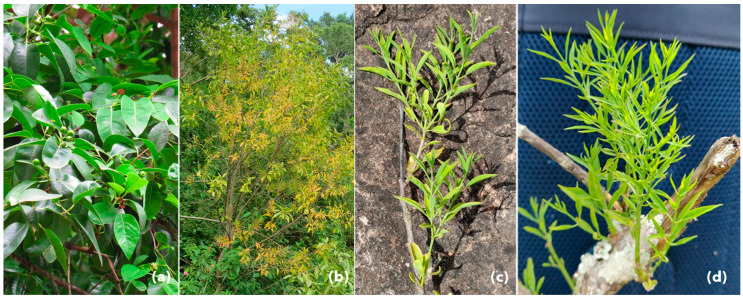
Healthy sandalwood tree bearing levees, inflorescence, and fruits (**a**) and phytoplasma infected sandalwood (*Santalum album* L.) tree showing typical symptoms of shoot spike, little leaf, and discoloration of leaves (**b**); sandalwood branches and twigs showing typical symptoms of SSD (**c**,**d**).

**Figure 4 biology-11-01494-f004:**
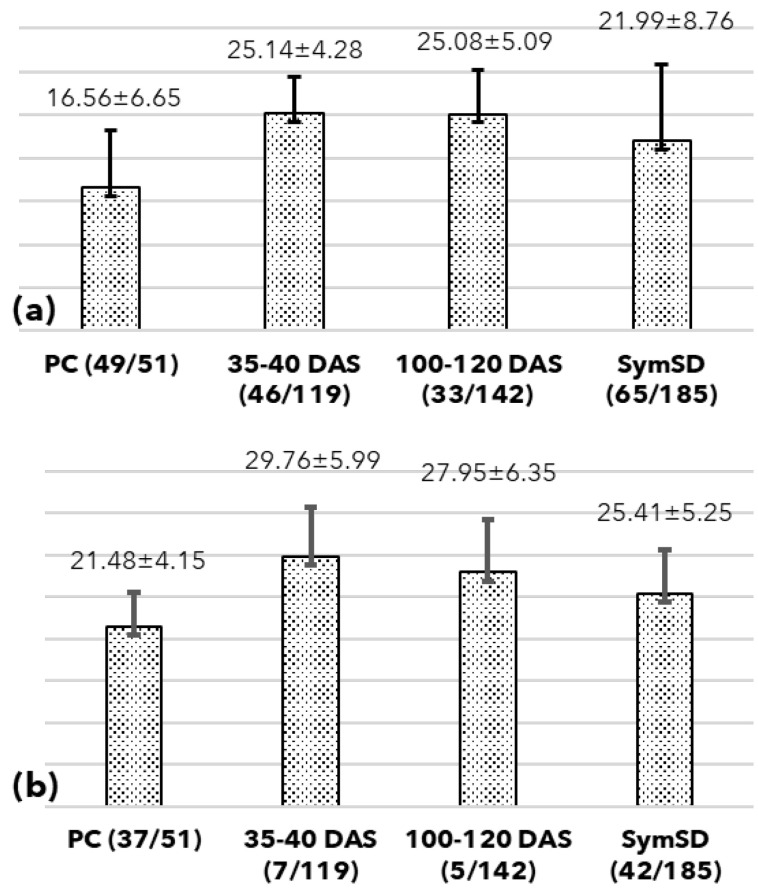
Range of Ct Values in samples screened using nested qPCR (**a**) and qPCR using genomic DNA template (**b**). Range of Ct values in positive control samples (PC), Seedlings collected at 35–40 DAS, Seedlings collected at 100–120 DAS, and seeds collected from SSD symptomatic sandalwood trees (SymSD). The figures in the parenthesis indicate the number of samples that tested positive for phytoplasma and the total number of samples screened.

**Figure 5 biology-11-01494-f005:**
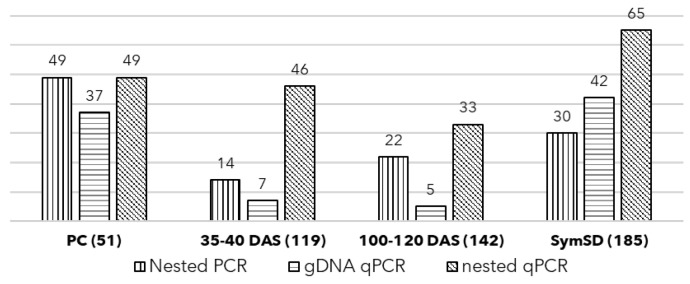
Phytoplasma positivity rate observed among three screening methods used; end-point nested PCR, qPCR using gDNA as a template, and nested qPCR using PCR product of P1/P7 as a template. Positivity rate observed in positive control samples (PC), seedlings collected at 35–40 DAS, seedlings collected at 100–120 DAS, and seeds collected from SSD symptomatic sandalwood trees (SymSD). The figures in the parenthesis indicate the number of samples screened for phytoplasma, and the figures on the top of the bar indicate the number of samples that tested positive for phytoplasma.

**Figure 6 biology-11-01494-f006:**
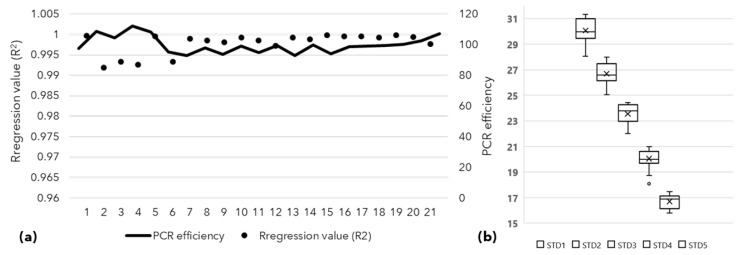
The PCR efficiency and regression (R^2^) values from nested qPCR assays distributed in 21 qPCR plates, ranged from 90 to 110 and close to 0.99, respectively (**a**); the graph showing mean and standard deviations of Ct values derived from serially diluted, (10^–3^ to 10^–7^; STD 1 to STD 5), purified, quantified 75 bp amplicon used to plot the standard curve across 21 qPCR plates (**b**).

**Figure 7 biology-11-01494-f007:**
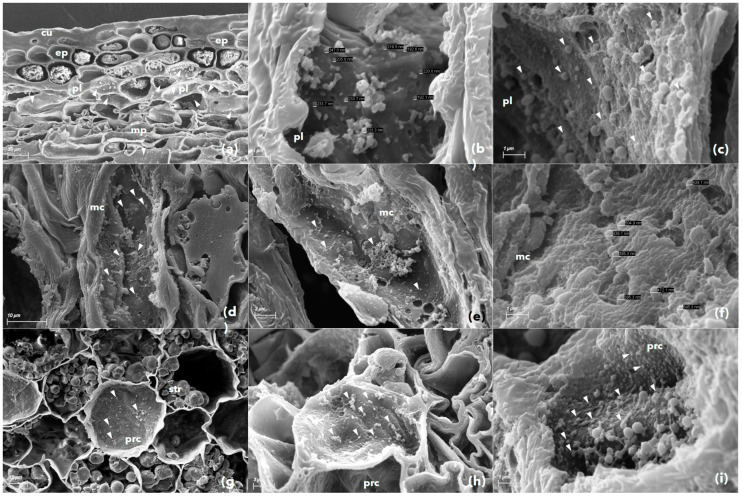
Visualization of phytoplasma cells in various tissues of sandalwood. SSD phytoplasma cells were observed in symptomatic sandalwood leaf tissue (**a**–**c**), in fruit mesocarp (**d**–**f**), and in hypocotyl tissue of sandalwood seedling (**g**–**i**). The location of the phytoplasma cells is indicated by the solid white triangle and the scale bars (**b**,**f**). cu, cuticle; ep, epidermal cells; pl, palisade cells; mp, mesophyll; ms, mesocarp and prc, parenchyma.

**Figure 8 biology-11-01494-f008:**
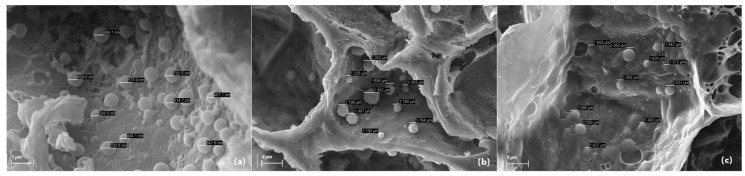
Scanning Electron Micrograph showing unknown bacterial endophytic cells in sandalwood seed (**a**,**b**) and hypocotyl tissue of seedling (**c**). The location of the bacterial cells is indicated by the scale bars.

## Data Availability

The GenBank/EMBL/DDBJ accession numbers for the reference 16S rRNA gene sequences of ‘*Candidatus* Phytoplasma’ were OM855930 to OM855981. The GenBank/EMBL/DDBJ accession numbers for bacterial and fungal strains are OM838448 to OM838470 and OM838424 to OM838447, respectively.

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
