# Peer review of "Nested Real-Time PCR Assessment of Vertical Transmission of Sandalwood Spike Phytoplasma (‘Ca. Phytoplasma asteris’)"

_biology, 2022, doi:10.3390/biology11101494_

Round 1

Reviewer 1 Report

The manuscript entitled “Nested Real-Time PCR Assessment of Vertical Transmission of Sandalwood Spike Phytoplasma ('Ca. Phytoplasma asteris')” represents a novel contribution to the understanding of epidemiology of SDS (Sandalwood Spike disease) by assessing possibility of phytoplasma transmission through seeds. Although the identification of phytoplasma in seeds and young seedlings is difficult due to the low concentration of the pathogen and requires nested PCR (end-point or real-time) which is as method questionable due to the high possibility of cross-contamination, it seems that authors have performed all experiments adequately and interpreted them rigorously. The results are very important not only for SDS epidemiology and disease control, but for overall understanding of phytoplasma transmission routes.

The manuscript is well written, methods are adequately chosen and results are well presented and discussed.

I would recommend changing the title since the Nested Real-Time PCR is not the only method used for assessment of vertical transmission of phytoplasma. I would suggest the following title: “Molecular Assessment of Vertical Transmission of Sandalwood Spike Phytoplasma ('Ca. Phytoplasma asteris')”.

Also there are some minor issues that need to be explained. These are as following:

Page 6, lines 252, 265 and 269: Please add Ct value and a standard deviation for qPCR using genomic DNA as template.

Page 6, lines 268-270: Please precisely indicate if the PCR positive plants (5/142 and 33/142) were symptomatic or asymptomatic.

It would be good, if possible, to screen for other pathogens also the seedlings which are asymptomatic as well as the seedlings which are positive for phytoplasma. These would serve as a negative and positive control for presented findings of other pathogens in symptomatic plants.

Author Response

The detailed point-to-point reply is enclosed in the PDF file attached.

Reviewer 2 Report

Authors have written the manuscript well. They have explained all the methods and results appropriately. At some points, there was little confusion about the samples used. Please find the minor corrections/suggestions described in the edited version of the manuscript. Authors are advised to revise the manuscript before accepting the manuscript for publication.

Author Response

(The authors gave the same response as above.)

Reviewer 3 Report

My comments can be found in the attached manuscript.

Author Response

(The authors gave the same response as above.)

Reviewer 4 Report

The authors in this study tested nested qPCR in the detection of Phytoplasma in Sandal wood. The data show that nested PCR is much better than pcr, however the authors missed to clearly show the results that indicate their findings. 

L16 what means distressful

L29 "fear"

L256 not clear where 900 comes from

L279 replace "figures" by numbers

The ms has too much abbreviation that are difficult to follow, the figures are not indicated in the text and it is not clear that is the difference between fig D and E. Further, the authors claim that other bacterial and fungal pathogens were isolated, by did not experimental provided proofs!  

Author Response

(The authors gave the same response as above.)

Round 2

Reviewer 3 Report

My comments can be found in the attached MS.

Author Response

File "reply to reviewers' attached.

Reviewer 4 Report

I thank the authors for their reply - My point thought has not been addressed. 

The authors claimed the isolation of bacterial and fungal pathogens in section 3.4 and sequenced 16S for bacteria and ITS for fungi. I understand that indicated genus include maybe pathogen but any instances the authors provided proofs in the 3.4 results section. Also, either I am having a wrong version but these results were not discussed. To claim that these isolates are plant-pathogens, the authors have to show it empirically. 

Author Response

File 'reply to reviewers' attached
